# Pfs48/45 nanobodies block *Plasmodium falciparum* transmission

**Frankie M. T. Lyons**[1,2], **Jill Chmielewski**[1,2], **Mikha Gabriela**[1,2], **Li-Jin Chan**[1,2], **Joshua Tong**[1], **Amy Adair**[1], **Kathleen Zeglinski**[1,2], **Quentin Gouil**[1,2,3,4], **Melanie H. Dietrich**[1,2‡], **Wai-Hong Tham** [1,2,5‡*]

1 The Walter and Eliza Hall Institute of Medical Research, Parkville, Victoria, Australia, 2 Department of Medical Biology, The University of Melbourne, Parkville, Victoria, Australia, 3 Olivia Newton-John Cancer Research Institute, Heidelberg, Victoria, Australia, 4 School of Cancer Medicine, La Trobe University, Bundoora, Victoria, Australia, 5 Research School of Biology, The Australian National University, Canberra, ACT, Australia

‡ Joint Senior Authors
* tham@wehi.edu.au

## Abstract

Malaria parasite fertilisation occurs within the *Anopheles* mosquito midgut. Interventions that inhibit parasite fertilisation prevent ongoing transmission and are important for malaria elimination efforts. Pfs48/45 and Pfs230 are two leading transmission-blocking vaccine candidates. Both proteins form a complex on the surface of sexual stage parasites and are essential for male fertility. Here we have identified nanobodies against Pfs48/45 that recognise gametocytes and have strong transmission-reducing activity. The crystal structure of our most potent nanobody in complex with Pfs48/45 reveals it binds a distinct epitope to TB31F, a leading transmission-blocking monoclonal antibody but to similar epitopes as RUPA-44 and RUPA-117. These results demonstrate the potential of nanobodies as a versatile antibody format that can reduce malaria transmission.

### Author summary

Malaria is spread when an infected *Anopheles* mosquito bites a human. Within the female *Anopheles* mosquito, malaria parasite fertilisation occurs in the mosquito midgut. If you inhibit parasite fertilisation in the mosquito, you can prevent onward transmission of the malaria parasite from mosquito to humans. Transmission blocking vaccines work by stopping parasite fertilisation and development in the mosquito and are key for malaria elimination by preventing community spread. Pfs48/45 and Pfs230 are two leading transmission-blocking vaccine candidates, and both are critical for male fertility. Here we describe the generation of nanobodies that target Pfs48/45. We show that when nanobodies against Pfs48/45 were added to infected blood meals for *Anopheles* mosquitoes,

**Data availability statement:** Coordinates and structure factor files of the Pfs48/45-nanobody B2 crystal structure have been deposited in the Protein Data Bank under PDB ID 9OBN. The NGS data has been deposited to the European Nucleotide Archive (ENA) under accession number PRJEB88774.

**Funding:** This work was supported by National Health and Medical Research Council of Australia (GNT2016908 and APP2001385 to W-H.T). The authors acknowledge the Victorian State Government Operational Infrastructure Support and Australian Government NHMRC IRIISS. The funders had no role in study design, data collection and analysis, decision to publish, or preparation of the manuscript.

**Competing interests:** The authors have declared that no competing interests exists.

the nanobodies significantly reduced parasite transmission. Our work demonstrates the potential of nanobodies as a versatile antibody format that can reduce malaria transmission.

## Introduction

Transmission-blocking interventions target malaria parasites in the mosquito host to prevent their onward transmission to humans. Malaria parasites are transmitted when a female *Anopheles* mosquito takes a blood meal from an infected human. Sexual stage malaria parasites called gametocytes are taken up and enter the mosquito midgut. After emerging from red blood cells, female gametocytes form macrogametes and male gametocytes undergo exflagellation to release microgametes. During fertilisation, microgametes attach to macrogametes and their membranes fuse. This leads to the development of zygotes, which elongate to form highly motile ookinetes. Ookinetes traverse the midgut epithelium and form oocysts on the outer wall of the midgut. Oocysts mature and rupture to release sporozoites, which travel to the mosquito salivary glands where they are ready to infect another human host. The oocyst stage represents a major bottleneck in the malaria parasite life cycle where parasite numbers are very low, providing an opportunity to eliminate infection from the mosquito completely and effectively inhibit malaria transmission [1].

Pfs48/45 is one of the leading candidates for transmission blocking vaccines and a major target of monoclonal antibodies to stop transmission. It is a member of the 6-cysteine protein family, whose members share a common 6-cysteine domain fold and play important roles throughout the parasite life cycle [2–4]. Pfs48/45 has three 6-cysteine domains; the N-terminal Domain 1 (D1), central Domain 2 (D2) and C-terminal Domain 3 (D3) [5,6]. Crystal structures and cryo-EM structures of Pfs48/45 indicate that the three 6-cysteine domains can adopt a planar triangular arrangement on the parasite surface as well as an extended conformation [7,8]. Pfs48/45 is expressed on the surface of male and female gametocytes and gametes [9–15] and is predicted to be tethered to the parasite membrane via its glycosylphosphatidylinositol (GPI) anchor [11,16]. Pfs48/45 interacts with Pfs230, another 6-cysteine protein and major transmission-blocking target, and is thought to be responsible for the localisation of Pfs230 at the parasite surface [17–19]. P48/45 appears to be important for gamete attachment and male fertility, with male knockouts in *P. falciparum* and *P. berghei* unable to fertilise female gametes [12,14,20].

Sera from individuals naturally infected with *P. falciparum* contains Pfs48/45-specific antibodies that are correlated with transmission-blocking capability [21–26]. While initial efforts were focussed on D3, transmission-blocking epitopes have since been found to span all three 6-cysteine domains and anti-Pfs48/45 monoclonal antibodies (mAbs) targeting D1 and D3 have potent transmission-blocking activity [7,27–33]. Crystal structures and cryo-electron microscopy (cryo-EM) structures for eleven transmission blocking mAbs in complex with Pfs48/45 are available, of which seven bind to D3, two to D2 and two to D1 [7,8,30–32,34,35]. Antibodies 32F3,

85RF45.1 and the humanised version of 85RF45.1, TB31F, have overlapping epitopes on D3 and are well-characterised to have potent transmission-blocking activity [7,30,31]. More recently, four potent transmission-blocking mAbs isolated from naturally infected donors have been structurally characterised [32,34]. RUPA-47 and RUPA-29 recognise the same region of D3 as previously characterised antibodies, while RUPA-44 and RUPA-117 were found to recognise a different region of Pfs48/45 D3 [32,34]. 10D8 binds to D2 and has moderate transmission-blocking activity [7,30].

Anti-Pfs48/45 monoclonal antibodies are being pursued as an intervention strategy to stop transmission. A phase I trial of TB31F as a malaria prophylactic in malaria-naïve participants found it to be well tolerated and potent, with the potential for a single dose to provide protection for an entire malaria season [36,37]. Targeting of multiple antigens or epitopes could reduce the risk of escape mutants and enhance inhibitory activity through additive or synergistic effects [38,39]. This has been observed for Pfs48/45 and Pfs230, where mice immunised with a chimeric Pfs230-Pfs48/45 vaccine had transmission-blocking antibody responses three-fold higher than those vaccinated with the single antigens alone [40]. Moreover, cocktails of mAbs targeting different epitopes of Pfs230 Domain 1 (D1) or targeting both Pfs230 D1 and Pfs48/45 showed significantly increased transmission-blocking activity compared to individual antibodies [41].

Nanobodies are the smallest naturally occurring antigen binding domain. This provides several advantages compared to conventional antibodies such as a smaller size (~15 kDa), increased stability across pH and temperature ranges and cost-effective expression in bacterial systems. In addition, nanobodies are modular single domain antibodies that can be easily linked together to generate diverse antibody formats such as biparatopic or bispecific entities. Nanobodies have been successfully isolated against several *P. falciparum* 6-cysteine proteins, such as Pf12p, Pf12, Pf41 and Pfs230, and these nanobodies show high affinity and specificity against their respective antigens [42–44]. Furthermore, nanobodies targeting Pfs230 have previously been shown to have transmission-blocking activity [42].

Here we aimed to investigate the potential of nanobodies targeting Pfs48/45 as a transmission-blocking intervention. We isolated three anti-Pfs48/45 nanobodies that bound Pfs48/45 D3 with nanomolar affinity. All three nanobodies showed transmission-reducing activity, with one showing near complete blocking of malaria transmission at 100 µg/mL. The crystal structure of this nanobody in complex with Pfs48/45 D3 reveals its binding epitope is distinct to that of TB31F.

## Results

### Isolation of specific nanobodies against Pfs48/45

To identify nanobodies against Pfs48/45, we immunised an alpaca with recombinant Pfs48/45 D3 that had been deglycosylated using PNGase F (Figs 1A and S1A) and generated an associated nanobody phage library. Following two rounds of phage display, we observed 54% of 94 phage supernatants bound Pfs48/45 by ELISA (S1B Fig). Sanger sequencing identified 10 distinct nanobody clonal groups between which the sequence of complementarity determining region 3 (CDR3) varied by three or more amino acids (Figs 1B and S1C and S1 Table). The length of the nanobody CDR3 sequences ranged between 15 and 24 residues (Fig 1B and S1 Table). Sequences of all 10 nanobodies were present in next-generation sequencing analysis of the nanobody library after two rounds of phage display, with varying levels of abundance (Figs 1C and S1D). A representative nanobody from each clonal group (A4, B2, C1, C3, C7, D3, D4, F6, G3 and H4) was expressed and purified as an Fc-fusion protein (~80 kDa) (S2 Fig).

We tested whether nanobody-Fcs were able to detect recombinant Pfs48/45 D3 by western blotting. Deglycosylated Pfs48/45 D3 with a molecular weight of 19 kDa was used. WNb7-Fc, a SARS-CoV-2-specific nanobody [45], and PBS were used as negative controls. Anti-Pfs48/45 antibody TB31F [31] was used as a positive control. All purified nanobody-Fcs were able to detect recombinant Pfs48/45 D3 under non-reducing conditions by western blotting (Fig 1D). Under reducing conditions, only A4-Fc, B2-Fc, C1-Fc, C3-Fc, D4-Fc and G3-Fc showed reactivity to reduced Pfs48/45 D3 (Fig 1D). As expected, neither WNb7-Fc or PBS showed reactivity for Pfs48/45 D3 under reducing or non-reducing conditions (Fig 1D). TB31F showed reactivity to Pfs48/45 D3 under non-reducing conditions only (Fig 1D), as previously described [28,30,32].

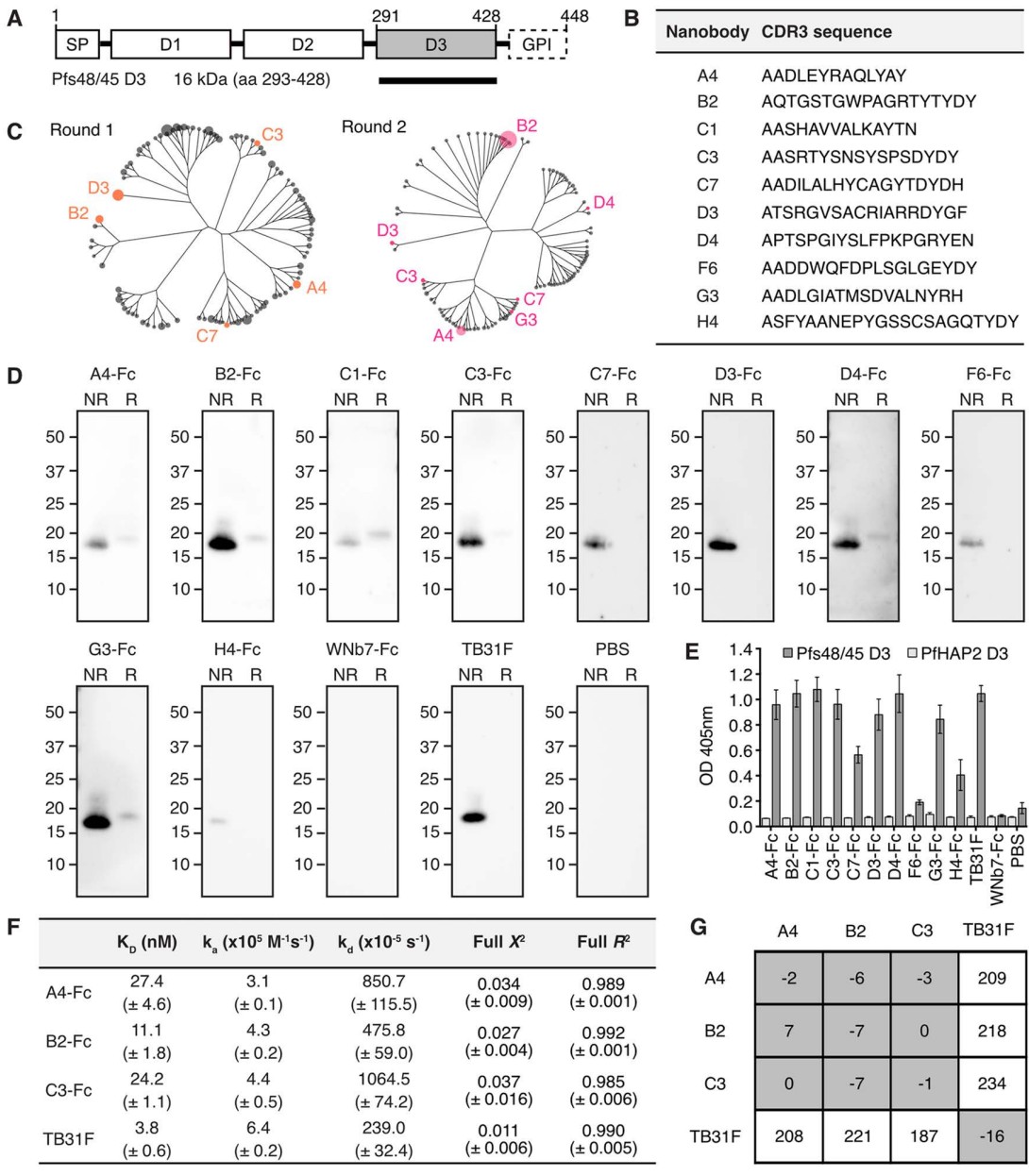

**Fig 1. Pfs48/45 D3-specific nanobodies. (A)** Schematic of Pfs48/45 and domain 3 (D3) recombinant construct (black bar). The signal peptide (SP), putative glycosylphosphatidylinositol (GPI) anchor and three 6-cysteine domains of Pfs48/45 are indicated. **(B)** Complementarity determining region 3 (CDR3) sequence of anti-Pfs48/45 nanobodies. **(C)** Cladogram of the 100 most abundant nanobodies from Next-Generation Sequencing (NGS) of round 1 and round 2 phage display selection libraries. Anti-Pfs48/45 nanobodies identified by Sanger sequencing are labelled. Tree tips are scaled relative to abundance (counts per million). **(D)** Western blots of nanobody-Fcs against recombinant Pfs48/45 D3. Non-reduced (NR) and reduced **(R)** PNGase F-deglycosylated Pfs48/45 D3 was separated by SDS-PAGE and probed with anti-Pfs48/45 nanobody-Fcs, an irrelevant nanobody-Fc (WNb7-Fc), anti-Pfs48/45 monoclonal antibody TB31F or PBS. HRP-conjugated anti-human IgG was used for detection. Molecular mass markers in kDa are indicated on the left-hand side of each blot. **(E)** ELISA of nanobody-Fcs against Pfs48/45. Microtiter wells were coated with recombinant Pfs48/45 D3 or PfHAP2 D3 and probed with anti-Pfs48/45 nanobody-Fcs, WNb7-Fc, TB31F or PBS. Bound nanobodies were detected with HRP-conjugated anti-human IgG. Error bars represent standard deviation. **(F)** Nanobody-Fc and TB31F affinities for Pfs48/45 D3 determined by bio-layer interferometry (BLI). Mean affinities ($K_D$), association rates ($k_a$), dissociation rates ($k_d$), $X^2$ and $R^2$ values of two independent experiments are given with the standard deviation in brackets. **(G)** Epitope competition experiments using BLI. Nanobodies or TB31F Fab along the top were pre-incubated with Pfs48/45 D3 at a 10:1 molar ratio. Nanobodies or TB31F indicated on the left were immobilised on the sensor and dipped into the pre-incubated solutions. The percentage binding of antigen pre-incubated with nanobody/Fab was calculated relative to antigen binding alone, which was assigned to 100%. The grey and white boxes represent competition and no competition, respectively.

We examined whether the nanobody-Fcs could bind Pfs48/45 D3 using ELISA. As a negative control, we used recombinant PfHAP2 Domain 3 (PfHAP2 D3), a *P. falciparum* sexual-stage antigen important for gamete membrane fusion [46,47]. All 10 nanobody-Fcs showed reactivity against Pfs48/45 D3 but not against PfHAP2 D3 (Fig 1E). As expected, TB31F showed reactivity to Pfs48/45 D3 but not to PfHAP2 D3, and negative controls WNb7-Fc and PBS showed no reactivity to either antigen (Fig 1E). These results demonstrate that the 10 nanobodies have specificity for Pfs48/45 D3.

Binding affinities of the nanobodies for Pfs48/45 D3 were determined using bio-layer interferometry (BLI). Binding affinities in the nanomolar range were observed for three nanobodies, namely A4-Fc, B2-Fc and C3-Fc, with $K_D$ values of 27.4, 11.1 and 24.2 nM, respectively (Fig 1F). While we could detect binding for C1-Fc, D3-Fc, D4-Fc and G3-Fc to Pfs48/45 D3, we were unable to fit the binding curves using a 1:1 Langmuir binding model (S3 Fig). Using BLI, no binding was detected for nanobodies C7-Fc, F6-Fc, H4-Fc or negative control WNb7-Fc (S3 Fig). TB31F bound to Pfs48/45 with low nanomolar affinity of 3.8 nM, as previously reported [31].

We used epitope binning to determine whether the higher affinity nanobodies A4, B2 and C3 bound to similar or distinct epitopes. Monomeric nanobodies (S2 Fig) or TB31F were immobilised on the sensor and dipped into a solution of Pfs48/45 D3 pre-incubated with monomeric nanobodies or TB31F Fab. We observed that all three nanobodies compete with each other for binding to Pfs48/45 D3 but do not compete with TB31F (Fig 1G). These results indicate that the A4, B2 and C3 nanobodies have overlapping epitopes but bind a different region of D3 to the potent transmission-blocking antibody TB31F.

## Anti-Pfs48/45 D3 nanobodies have transmission-reducing activity

We used western blotting to determine whether nanobody-Fcs were able to detect full-length Pfs48/45 from *P. falciparum* NF54 iGP2 [48] stage V gametocyte lysate. Pfs48/45 migrates as a doublet of 48 and 45 kDa under non-reducing conditions, and as a single 58 kDa band under reducing conditions [10,49]. Neither WNb7-Fc or PBS showed reactivity for Pfs48/45 under reducing or non-reducing conditions (Fig 2A), although we did observe bands at ~55 kDa under reducing conditions and ~170 kDa under non-reducing conditions due to non-specific binding of the secondary antibody (Fig 2A). The presence of the 55 kDa band potentially impedes detection of Pfs48/45 binding in reducing conditions, but none of the nanobody-Fcs showed any detectable binding to reduced Pfs48/45 (Fig 2A). All 10 nanobody-Fcs recognised Pfs48/45 under non-reducing conditions, as did TB31F (Fig 2A). There were additional bands detected for C1-Fc and G3-Fc under non-reducing and reducing conditions that were not present in the negative controls, perhaps suggesting off target recognition of another protein in *P. falciparum* (Fig 2A).

We conducted an immunofluorescence assay with fixed stage V gametocytes using TB31F directly conjugated to Alexa Fluor 647 and nanobody-Fcs detected with anti-human IgG conjugated to Alexa Fluor 488. For this experiment, we only examined the staining of the three nanobodies that bound to Pfs48/45 D3 with high affinities by BLI; A4-Fc, B2-Fc and C3-Fc. The staining of all three anti-Pfs48/45 nanobody-Fcs colocalises with that of TB31F (Fig 2B). As expected, negative control WNb7-Fc did not show any detectable staining of gametocytes (Fig 2B).

We tested the transmission-reducing activity of A4-Fc, B2-Fc and C3-Fc by standard membrane feeding assay (SMFA). At 200 µg/mL, TB31F demonstrated complete blocking of parasite transmission, with no oocysts present in both experiment 1 and experiment 2 (Fig 2C), as previously described [31,32]. In experiment 1, our PBS and WNb7-Fc controls had mean oocyst counts of 35.7 and 21.9, respectively. All three nanobody-Fcs at 100 µg/mL reduced oocyst numbers in experiment 1: A4-Fc had mean oocyst counts of 3.9, B2-Fc had mean oocyst counts of 0.4 and C3-Fc had mean oocyst counts of 7.3 (Fig 2C). For experiment 2, mean oocyst counts were 6.6 and 6.6 for PBS and WNb7-Fc, respectively (Fig 2C). All three nanobody-Fcs at 100 µg/mL reduced oocyst numbers: A4-Fc had mean oocyst counts of 0.9, B2-Fc had mean oocyst counts 0.1 and C3-Fc had mean oocyst counts of 0.3 (Fig 2C). Collectively, these results show that A4-Fc and C3-Fc had significant transmission-reducing activity of 86–89% and 79–95%, respectively (Fig 2C). B2-Fc was the most potent nanobody, with a transmission-reducing activity of 99%, and reduced infection prevalence as compared to PBS (5–26% vs 89–100%) (Fig 2C).

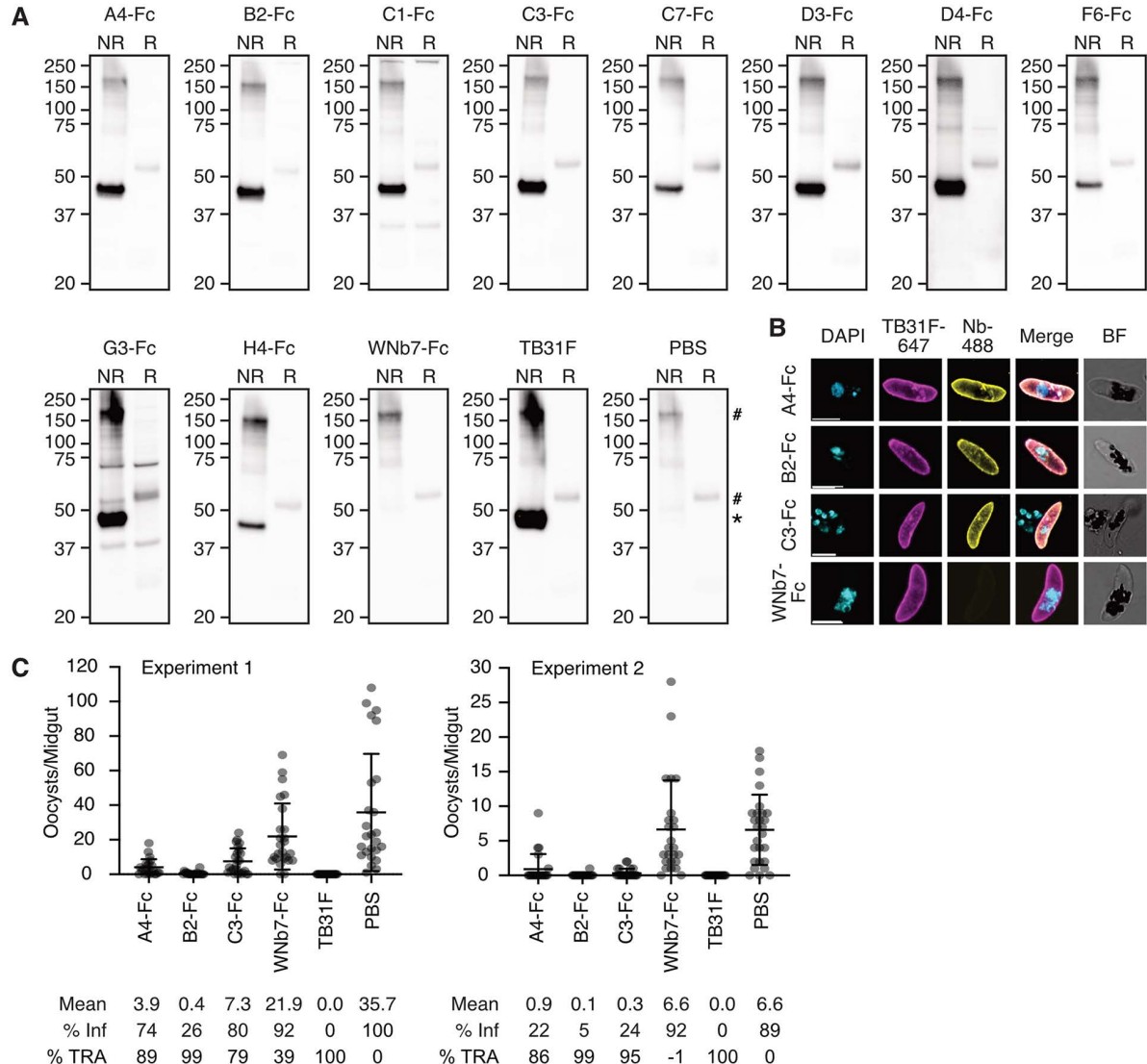

**Fig 2. Anti-Pfs48/45 nanobodies block malaria transmission. (A)** Western blots of nanobody-Fcs against Pfs48/45 from *P. falciparum* stage V gametocyte lysate. Non-reduced (NR) and reduced (R) lysate was separated by SDS-PAGE and probed with anti-Pfs48/45 nanobody-Fcs, an irrelevant nanobody-Fc (WNb7-Fc), anti-Pfs48/45 monoclonal antibody TB31F or PBS. HRP-conjugated anti-human IgG was used for detection. Molecular weight markers in kDa are indicated on the left-hand side of each blot. * = Pfs48/45 band, # = non-specific band. **(B)** Immunofluorescence assay of stage V gametocytes using the indicated nanobody-Fc detected with goat anti-human IgG conjugated to Alexa Fluor 488 (Nb-488, yellow) and anti-Pfs48/45 TB31F antibody conjugated to Alexa 647 (magenta). Parasite nuclei are stained with DAPI (cyan) and merged and brightfield (BF) images are shown. Scale bar = 5 μm. **(C)** Data from two independent standard membrane feeding assay (SMFA) experiments testing the transmission-blocking activity of anti-Pfs48/45 nanobody-Fcs. *Anopheles stephensi* mosquitoes were dissected 7-9 days post feeding with stage V NF54 iGP2 gametocytes and oocyst numbers quantified. PBS and WNb7-Fc were included as negative controls and TB31F as a positive control. Blood meals included heat-inactivated human serum, 100 μg/mL of nanobody-Fcs or 200 μg/mL TB31F in PBS or 10 μL PBS. Number of oocysts per dissected midgut are plotted. Bars show means with standard deviation. % Inf = % of mosquitoes infected; %TRA = transmission-reducing activity relative to the PBS control.

## Binding epitope of transmission-blocking nanobody B2 on Pfs48/45 D3

To elucidate the binding epitope of our most potent transmission-blocking nanobody, we determined the crystal structure of nanobody B2 bound to Pfs48/45 D3 to a resolution of 2.5 Å (Fig 3A and Table 1). The interface area is 590 Å, with the

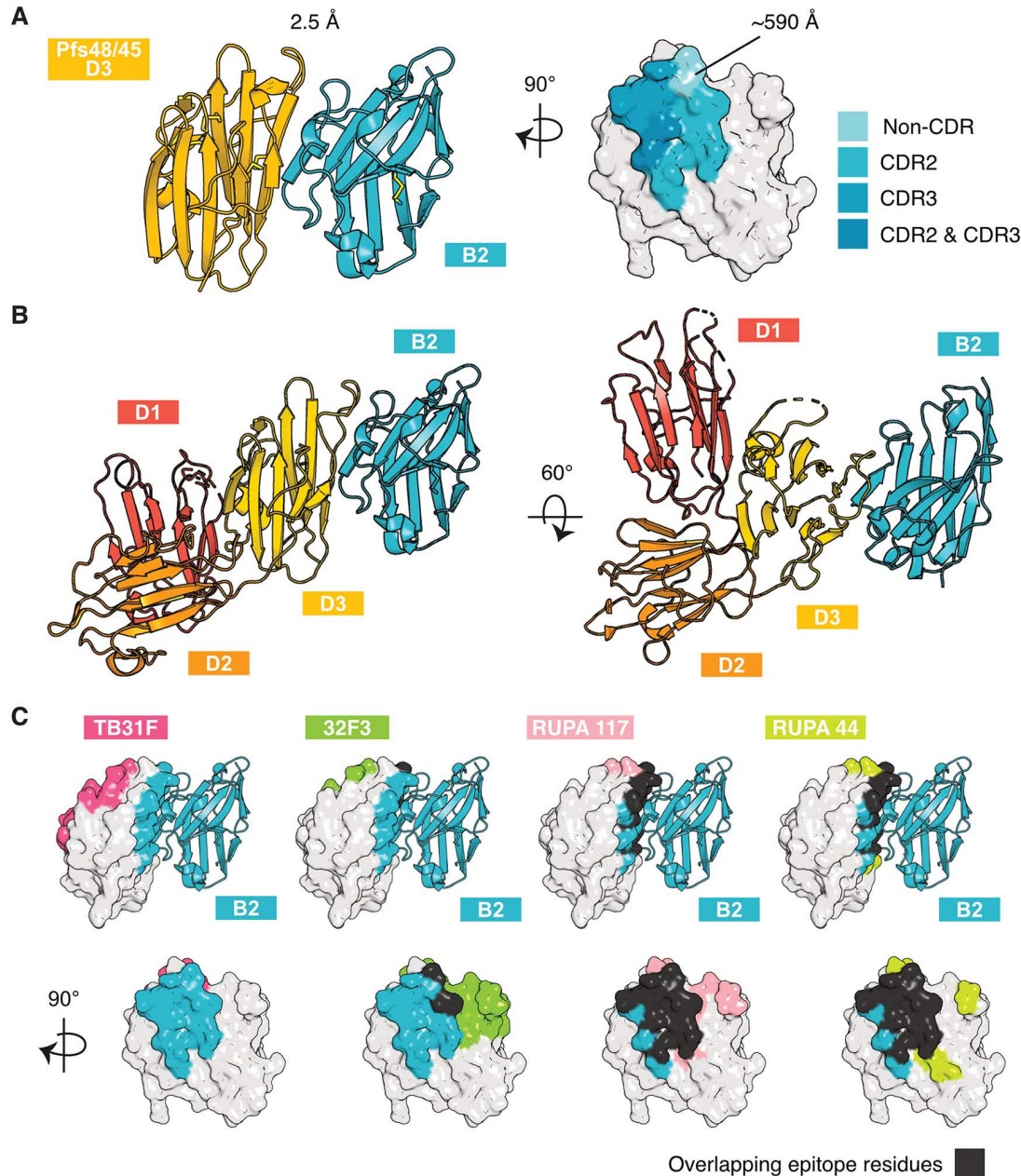

**Fig 3. Pfs48/45 D3 in complex with transmission-blocking nanobody. (A)** Crystal structure of Pfs48/45 D3 in complex with nanobody B2. Left: Ribbon representation of Pfs48/45 D3 in yellow and nanobody B2 in blue. Right: Surface representation of Pfs48/45 D3 showing the interface with nanobody B2. Residues of Pfs48/45 D3 within 5 Å of B2 are coloured, with contacts of complementarity determining regions (CDRs) 2 and 3 highlighted. Interface surface area calculated using PISA v2.1.0. **(B)** Binding site of nanobody B2 on Pfs48/45 D1D2D3. The structure of all three domains of Pfs48/45 from PDB ID 7ZXG is aligned to nanobody B2 via D3 of Pfs48/45. **(C)** Pfs48/45 with binding epitopes of nanobody B2 and transmission-blocking antibodies TB31F (PDB ID 6E63), 32F3 (PDB ID 7ZWI), RUPA-117 (PDB ID 7UNB) and RUPA-44 (PDB ID 7UXL), which have previously been crystallised in complex with Pfs48/45. Epitope residues within 5 Å of antibodies are coloured. B2 epitope residues are coloured in blue, and overlapping epitope residues in black.

**Table 1. Data collection and refinement statistics for the Pfs48/45 D3-B2 complex.**

| | Pfs48/45 D3-B2 (PDB 9OBN) |
|---|---|
| Data collection statistics | |
| Wavelength (Å) | 0.953732 |
| Space group | $P22_12_1$ |
| Cell axes (Å) (a, b, c) | 47.70, 88.55, 281.84 |
| Cell angles (°) (α, γ, β) | 90, 90, 90 |
| Resolution range (Å) | 47.54-2.50 (2.65-2.50)* |
| Completeness (%) | 99.8 (99.8) |
| Total no. of reflections | 445756 (58212) |
| Unique reflections | 42503 (6759) |
| Redundancy | 10.5 (8.6) |
| $R_{meas}$ (%) | 26.1 (147.5) |
| $CC_{1/2}$ (%) | 99.3 (68.7) |
| I/σ | 8.38 (1.82) |
| Wilson B (Å$^2$) | 46.54 |
| Refinement statistics | |
| $R_{work}/R_{free}$ (%) | 20.8/ 25.4 |
| No. of atoms | |
| Protein | 7879 |
| Water | 383 |
| Small molecules | 40 |
| B factors (Å$^2$) | |
| Chain A | 42.3 |
| Chain B | 49.3 |
| Chain C | 43.3 |
| Chain D | 61.8 |
| Chain E | 38.2 |
| Chain F | 43.2 |
| Chain G | 43.2 |
| Chain H | 42.2 |
| Water | 42.9 |
| Small molecules | 54.2 |
| R.m.s. deviations | |
| Bond lengths (Å) | 0.004 |
| Bond angles (°) | 0.652 |
| Validation | |
| Ramachandran outliers (%) | 0.00 |
| Ramachandran favored (%) | 97.60 |
| Rotamer outliers (%) | 0.37 |
| C-beta outliers | 0.00 |
| MolProbity score | 1.32 |

* The values in parentheses represent the highest-resolution shell.

interaction mediated by the CDR2 and CDR3 loops of B2 (Fig 3A). Residues of the CDR2 loop form 11 hydrogen bonds with residues D312, S322, H324, S326 and N328 of Pfs48/45 D3 (S2 Table). Residue R108 of the CDR3 loop forms a salt bridge with residue D320 of Pfs48/45 D3 (S2 Table).

We mapped the binding site of nanobody B2 to the crystal structure of the Pfs48/45 ectodomain (Fig 3B) (PDB ID 7ZXG) [7]. We observe that Pfs48/45 D1 and D2 are not predicted to occlude binding of nanobody B2 to D3. In addition, we compared the binding epitope of B2 with those of previously published anti-Pfs48/45 D3 antibodies TB31F (PDB ID 6E63), 32F3 (PDB ID 7ZWI), RUPA-117 (PDB ID 7UNB) and RUPA-44 (PDB ID 7UXL) (Fig 3C) [7,31,32]. B2 binds a different face of Pfs48/45 D3 to TB31F (Fig 3C). This is consistent with the results obtained in our epitope binning experiments, where B2 does not compete for binding with TB31F (Fig 1G). The B2 epitope overlaps with that of 32F3 by two residues and with those of RUPA-117 and RUPA-44 by 13 residues (Fig 3C, bottom row).

## Discussion

To the best of our knowledge, we present the first panel of nanobodies against transmission-blocking vaccine candidate Pfs48/45. We identified three anti-Pfs48/45 nanobodies, A4-Fc, B2-Fc and C3-Fc, that bind with affinities in the nanomolar range and reduce *P. falciparum* transmission.

Nanobody B2 recognises a region of Pfs48/45 D3 [32] distinct to the epitope of leading transmission-blocking monoclonal TB31F. The binding epitope of B2 is highly similar to that of RUPA-44 and RUPA-117, two human mAbs with potent transmission-blocking activity [32]. Three single nucleotide polymorphisms (SNPs) were described in the epitopes of these mAbs, namely L314, D320 and S322, with frequencies of 28.2%, 0.007% and 39.0%, respectively [32,50]. None of these point mutations were found to significantly impact the binding of RUPA-44 or RUPA-117 [32]. Unlike in the interaction with RUPA-44 and RUPA-117, residue S322 is involved in hydrogen bonding with B2. As the hydrogen bond involves the backbone carbonyl oxygen of S322 and not the sidechain, a polymorphism at this site is unlikely to disrupt the hydrogen bond and is not expected to affect B2 binding.

There is increasing interest in administering antimalarial monoclonals to prevent malaria transmission in endemic areas, particularly to reduce cases in high transmission areas or seasons, suppress outbreaks in regions nearing elimination, and prevent the spread of drug-resistant strains. Anti-Pfs48/45 mAb TB31F has undergone Phase I clinical trial and was found to be safe and effective at blocking *P. falciparum* transmission [36]. In the highest dose group, serum maintained above 80% TRA for 84 days of follow-up, suggesting that a single dose may be effective at preventing transmission for an entire season [36], and modelling has suggested that annual administration of TB31F may be effective at reducing malaria cases in seasonal settings [37]. At ~15 kDa, monomeric nanobodies are rapidly cleared from the bloodstream by the kidneys. In this study, nanobodies were fused to a human Fc domain, increasing half-life by increasing their molecular weight above the 50–60 kDa threshold for glomerular filtration. For the development of nanobody prophylactics, modifications to the Fc region to extend half-life further [51] and humanisation of the nanobody backbone [52] would need to be considered.

An alternative transmission-blocking approach is to express the nanobodies within the mosquito midgut using transgenic approaches. Nanobodies are small, are highly stable and soluble, and can be easily engineered into multivalent constructs against multiple targets. These properties are advantageous for their use in novel vector-borne disease strategies. In the tsetse fly, symbiotic gut bacteria have been engineered to express anti-trypanosomal nanobodies, and were capable of significantly reducing parasite development in the midgut [53,54]. Vectors can also be engineered to express nanobodies directly. *Aedes aegypti* mosquitoes have been engineered to express single domain antibodies against Chikungunya virus, with transgenic mosquitoes showing significant reductions in viral replication and transmission potential [55]. Transgenic *Anopheles stephensi* mosquitoes were engineered to express single-chain antibodies against mosquito-stage antigens Pfs25, chitinase 1 or circumsporozoite protein [56]. All three transgenic mosquito lines inhibited parasite development and lowered infection levels significantly, but did not block transmission completely [56]. Transgenic mosquitoes

expressing single-chain version of antibody 2A10, which binds sporozoites of the malaria parasite *Plasmodium falciparum,* showed a reduction in transmission of *Plasmodium berghei* expressing the *P. falciparum* circumsporozoite protein to mice [57]. Expression of multivalent nanobodies in the *A. stephensi* vector is an attractive future transmission-blocking strategy, combining the favourable properties of nanobodies with the potent transmission-blocking potential of multivalent approaches.

Our nanobodies have extended the repertoire of transmission-reducing monoclonals targeting 6-cysteine transmission-blocking candidates. Nanobody design incorporating the structural basis for the interaction of Pfs48/45 and Pfs230 would assist in the development of potently inhibitory nanobodies for use as transmission-blocking prophylactics.

## Materials and methods

### Ethics statement

Immunisation and handling of the alpaca for scientific purposes was approved by Agriculture Victoria, Wildlife & Small Institutions Animal Ethics Committee, project approval No. 26–17. Use of human blood and serum from the Melbourne Red Cross LifeBlood for parasite culturing was approved by the Walter and Eliza Hall Institute (WEHI) Human Research Ethics Committee, project approval No. 86–17.

### Recombinant protein expression and purification

The Pfs48/45 D3 sequence was retrieved from PlasmoDB (PF3D7_1346700, amino acids K293-A428). Pfs48/45 D3 for alpaca immunisation was expressed in an insect cell expression system. DNA was codon optimised to *Spodoptera frugiperda* (*Sf*) and cloned into a modified form of baculovirus transfer vector pAcGP67-A. The Pfs48/45 sequence is in frame with the GP67-signal sequence, a C-terminal octa-histidine tag and AviTag, and a Tobacco Etch Virus (TEV) protease cleavage site. Pfs48/45 D3 was produced using Sf21 cells (Life Technologies) and Insect-XPRESS Protein-free Insect Cell Medium supplemented with L-glutamine (Lonza). A cell culture of ~$1.8 \times 10^6$ cells/mL was inoculated with the third passage stock of virus and incubated for three days at 28°C. Cells were separated from the supernatant by centrifugation at 13,000 x g for 20 min. A cOmplete EDTA-free protein inhibitor tablet (Roche) and 1 mL 200 mM phenylmethylsulfonyl fluoride (PMSF) was added per litre of culture. The supernatant was sterile filtered with a 0.45 µm filter and concentrated via tangential flow filtration using a 10 kDa molecular weight cut-off cassette (Millipore). Concentrated supernatant was sterile filtered with a 0.45 µm filter and dialysed into 30 mM Tris pH 7.5, 300 mM NaCl (buffer A). The dialysed sample was incubated with Ni-NTA resin (Qiagen) for 1 h at 4°C on a roller shaker. A gravity flow chromatography column was washed with 10 – 20 column volumes of buffer A followed by buffer A with stepwise increases in imidazole concentration. Pfs48/45 D3 eluted at imidazole concentrations of 70–200 mM. These fractions were concentrated for size exclusion chromatography (SEC) and applied to a Superdex 75 Increase 10/300 GL column (Cytiva) pre-equilibrated with 20 mM HEPES pH 7.5, 150 mM NaCl. Pfs48/45 D3 was then deglycosylated with PNGase F (NEB). Protein was diluted to 1 mg/mL in 1X Glyco-Buffer 2 (NEB) and 1 µL PNGase F was added per 100 µg protein. The reaction was allowed to proceed at 37°C overnight. The mixture was concentrated and Pfs48/45 D3 was separated by SEC using a Superdex 75 Increase 10/300 GL column (Cytiva) pre-equilibrated with 20 mM HEPES pH 7.5, 150 mM NaCl. Fractions containing deglycosylated Pfs48/45 D3 were pooled and concentrated for alpaca immunisation.

Stabilised Pfs48/45 D3 (D3.mAgE1, amino acids K291-A428 with mutations G397L, H308Y and I402V) [34] for nanobody characterisation and crystallisation was expressed in a transient mammalian expression system. The DNA sequence was optimised to *Homo sapiens* and included a C-terminal hexa-histidine tag and AviTag. Expi293 cells were transfected with D3.mAgE1 DNA using an ExpiFectamine 293 Transfection Kit (Gibco, Cat # A14526). Cells were incubated for three days at 37°C, 8% $CO_2$. The supernatant was separated by centrifugation at 4,000 x g for 20 min and sterile filtered with a 0.22 µm filter. Ni-NTA purification and SEC were performed as described above. Protein was deglycosylated with PNGase F as described above.

For crystallisation, the hexa-histidine tag and Avitag were removed by TEV protease cleavage. Protein was diluted to <0.3 mg/mL in 20 mM HEPES pH 7.5, 150 mM NaCl and 1 mg TEV protease was added per 10 mg protein. Cleavage was allowed to proceed at 4°C overnight. The cleavage mixture was loaded onto a 1 ml HisTrap Excel column (Cytiva) equilibrated in 20 mM HEPES pH 7.5, 150 mM NaCl and the cleaved protein was retrieved in the flow through.

PfHAP2 D3 (PF3D7_1014200, amino acids V501-N620) was expressed with a C-terminal hexa-histidine tag and Avitag. Expression in Sf21 cells and purification was performed as described above for Pfs48/45 D3. Pfs230 D1 (PF3D7_0209000, amino acids T587-G731) expression and purification was performed as previously described [42].

## Isolation of nanobodies against Pfs48/45

One alpaca was immunised six times with ~150 µg of recombinant Pfs48/45 D3 using GERBU FAMA (GERBU Biotechnik) as an adjuvant. Blood was collected three days after the last immunisation for the preparation of lymphocytes. Nanobody library construction was carried out according to established methods [42–44,58]. Briefly, alpaca lymphocyte mRNA was extracted and amplified by RT-PCR with specific primers to generate a cDNA library size of $10^5$ nanobodies with 85% correctly sized nanobody inserts. The library was cloned into a pMES4 phagemid vector, amplified in TG1 *E. coli* cells and subsequently infected with M13K07 helper phage (NEB, Cat # N0315) for phage expression. Phage display was performed as previously described [42–44,58] with two rounds of biopanning on 1 µg of immobilised Pfs48/45 D3. Positive clones were identified using ELISA and were sequenced and annotated using IMGT/V-QUEST [59] and aligned in Geneious Prime 2022.1.1 (https://www.geneious.com).

## Next-generation sequencing and analyses of nanobody phage libraries

Nanobodies were digested from the pMES4 phagemid vector using three different pairs of restriction enzymes to maintain sequence diversity and purified by gel extraction. Paired-end 2x300 bp sequencing libraries were prepared using the NEBNext Multiplex Oligos for Illumina (Cat # E7395) in a PCR-free manner according to the manufacturer's instructions and sequenced on an Illumina NextSeq 2000 instrument. Raw sequencing reads were trimmed to remove sequencing adapters with TrimGalore v0.6.7 (https://github.com/FelixKrueger/TrimGalore) and then merged using FLASH (v1.2.11) [60]. Annotation of nanobody sequences was performed using IgBLAST (v1.19.0) [61] with a reference database built from IMGT [62]. Nanobodies were collapsed into clones at the CDR3 level, and the counts of clones were normalised through conversion to counts per million (CPM). Cladograms were generated from generalised Levenshtein distances between CDR3s with the Neighbour-Joining method [63] from the phangorn (v2.12.1) [64] package and plotted using the ggtree (v3.14.0) [65] package in R (v4.4.1) [66].

## Nanobody and nanobody-Fc expression and purification

Nanobodies were expressed in *E. coli* WK6 cells. Bacteria were grown in Terrific Broth supplemented with 0.1% (w/v) glucose and 100 µg/mL carbenicillin at 37°C to an $OD_{600}$ of 0.7, induced with 1 mM IPTG and grown overnight at 28°C. Cell pellets were harvested and resuspended in 20% (w/v) sucrose, 10 mM imidazole pH 7.5, 150 mM NaCl PBS and incubated on ice for 15 min. EDTA pH 8.0 was added to a final concentration of 5 mM and incubated on ice for 20 min. After this incubation, 10 mM $MgCl_2$ was added, and periplasmic extracts were harvested by centrifugation at 4,000 x g for 1 hr. The supernatant was sterile filtered with a 0.22 µm filter and loaded onto a 1 ml HisTrap Excel column (Cytiva) equilibrated in 5 mM imidazole pH 7.5, 100 mM NaCl PBS. Nanobodies were eluted with 400 mM imidazole pH 7.5, 100 mM NaCl PBS, and subsequently concentrated and buffer exchanged into 20 mM HEPES pH 7.5, 150 mM NaCl.

Nanobody sequences were subcloned into a derivative of pHLSec containing the hinge and Fc region of human IgG1 using PstI and BstEII restriction sites. Nanobody-Fcs were expressed in Expi293 cells via transient transfection. The supernatant was harvested seven days after transfection and applied to 1mL HiTrap PrismA affinity columns (Cytiva) equilibrated in PBS. Nanobody-Fcs were eluted in 100 mM citric acid pH 3.0 and neutralised by the addition of 1 M

Tris-HCl pH 9.0. They were subsequently buffer exchanged into PBS and purified by SEC using a Superdex 200 Increase 10/300 GL column (Cytiva) equilibrated in 20 mM HEPES pH 7.5, 150 mM NaCl.

### Antibody expression and purification

The variable regions of the heavy and light chain of anti-Pfs48/45 mAb TB31F [31] were synthesised as gBlocks (IDT). The heavy chain sequence was subcloned into an AbVec-hIgG1 plasmid (AddGene) using AgeI and SalI restriction enzyme sites. The light chain sequence was subcloned into an AbVec-IgKappa plasmid (AddGene), using the AgeI and HindIII restriction enzyme sites to replace the IgKappa sequence with the TB31F $V_L$ and IgLambda sequence. TB31F was expressed in Expi293 cells via transient transfection using a 1:1 ratio of light and heavy chain plasmids and purified following the same protocol as described above for nanobody-Fcs. For epitope binning experiments, FabALACTICA (IgdE, Genovis) was used to generate TB31F Fab fragments. TB31F was diluted to 2 mg/mL in PBS and 50 μg enzyme was added per mg of protein. The cleavage was allowed to proceed for 72 hours at 37°C and then applied to a HiTrap PrismA affinity column (Cytiva) equilibrated in PBS. Flow through was collected and applied to a 1 mL HisTrap Excel column (Cytiva). The flow through containing TB31F Fab was collected and concentrated.

### Nanobody specificity using ELISA

Flat-bottomed 96-well MaxiSorp plates were coated with Pfs48/45 D3.mAgE1 diluted to a concentration of 125 nM in 50 μL PBS at room temperature (RT) for 1 h. All washes were done three times using PBS and 0.05% Tween (PBST) and all incubations were performed for 1 h at RT. Coated plates were washed and blocked by incubation with 10% skimmed milk in PBST. Plates were washed and then incubated with 125 nM nanobody-Fcs in 50 μL PBS. The plates were washed and incubated with horseradish peroxidase (HRP)-conjugated goat anti-human secondary antibody (Jackson ImmunoResearch, Cat # 109-035-088) diluted 1:5000. After a final wash, 50 μL of azino-bis-3-ethylbenthiazoline-6-sulfonic acid liquid substrate (ABTS; Sigma) was added and incubated at RT for ~20 min. 50 μL of 1% SDS was used to stop the reaction. Absorbance was read at 405 nm and all samples were tested in duplicate.

### Western blotting

For recombinant protein, 250 ng non-reduced and reduced Pfs48/45 D3.mAgE1 was loaded on a 4–12% Bis-Tris gel (NuPAGE) and SDS-PAGE was run in MES buffer at 200 V for 37 minutes. Separated proteins were transferred onto a nitrocellulose membrane using iBlot 2 (Invitrogen). The membrane was blocked with 10% milk in PBST for 1 h at RT and subsequently probed with 10 μg/mL nanobody-Fcs, 0.1 μg/mL TB31F or PBST for 1 h at RT. Following nanobody incubation, the membrane was washed for five minutes three times with 1X PBST and probed with 1:5000 HRP-conjugated anti-human secondary antibody for 1 h at RT. All nanobodies and antibodies were prepared in 1% milk PBST. Chemiluminescent detection was performed using SuperSignal West Pico PLUS Chemiluminescent Substrate (Thermo Scientific) and imaged with a ChemiDoc imaging system (Bio-Rad).

For western blotting with parasite lysate, synchronised stage V gametocyte cultures of *P. falciparum* NF54 iGP2 were used. Gametocyte pellets were treated with 0.03% saponin and diluted 1:20 in 2X non-reducing buffer, 10 μL of which was loaded onto a 4–12% Bis-Tris gel under non-reducing and reducing conditions. Blots were probed with 0.1 μg/mL antibodies or PBST for 1 h at RT. Transfer, antibody incubation and imaging was performed as described above for nanobody-Fc experiments.

### BLI for affinities

Affinity measurements were performed on the Octet RED96e (FortéBio) using anti-human IgG Fc Capture (AHC) sensor tips. All measurements were performed in kinetics buffer (PBS pH 7.4 supplemented with 0.1% (w/v) BSA and 0.05%

(v/v) TWEEN-20) at 25°C. After a 60 s baseline step, test nanobodies or antibodies at 5 µg/mL were loaded onto sensors, followed by another 60 s baseline step. Association of nanobodies or antibodies with a dilution series of antigen was measured over 180 s, followed by dissociation in kinetics buffer for 180 s. For nanobody-Fc experiments, nanobody-Fc association measurements were performed using a two-fold dilution series of Pfs48/45 D3.mAgE1 from 6 to 200 nM and TB31F using a two-fold dilution series of Pfs48/45 D3.mAgE1from 1.5 to 50 nM.

Sensor tips were regenerated using five cycles of 5 s in 100 mM glycine pH 1.5 and 5 s in kinetics buffer. Baseline drift was corrected by subtracting the response of a nanobody-loaded sensor incubated in kinetics buffer only. Curve fitting analysis was performed with Octet Data Analysis 10.0 software using a 1:1 Langmuir binding model to determine $K_D$ values and kinetic parameters. Curves that could not be fitted well (<0.98 $R^2$ and >0.5 $X^2$) were excluded from the analyses. The mean kinetic constants reported are the result of two independent experiments.

## BLI for epitope binning

For epitope binning experiments, 400 nM Pfs48/45 D3.mAgE1 was pre-incubated with each monomeric nanobody or TB31F Fab at a 10-fold molar excess on ice for 1 h. A 30 s baseline step was established between each step of the assay. NTA sensors were first loaded with 20 µg/mL of nanobody A4, 10 µg/mL of nanobody B2, 20 µg/mL of nanobody C3 or 30 µg/mL of nanobody WNb7 for 5 min. The sensor surface was then quenched by dipping into 8000 nM of an irrelevant nanobody for 5 min. To test for competition with TB31F, AHC sensors were loaded with 10 µg/mL of TB31F or WNb7-Fc for 5 min. Sensors were then dipped into the pre-incubated solutions of Pfs48/45 D3.mAgE1 with nanobody or Fab for five minutes. Loaded sensors were also dipped into Pfs48/45 D3.mAgE1 alone to determine the level of Pfs48/45 D3.mAgE1 binding to immobilised nanobody in the absence of other nanobodies. Percentage competition was calculated by dividing the response at 70 seconds of the premixed Pfs48/45 D3.mAgE1 and nanobody or Fab solution binding by the response at 70 seconds of Pfs48/45 D3.mAgE1 binding alone, multiplied by 100.

## Crystallisation and structure determination

TEV-cleaved and PNGase F-deglycosylated Pfs48/45 D3.mAgE1 was complexed with nanobody B2 at a molar ratio of 1:1.5 for 1 h on ice. Complexes were applied to a Superdex 75 Increase 10/300 GL column (Cytiva) pre-equilibrated in 20 mM HEPES pH 7.5, 150 mM NaCl and purified by SEC. Initial crystallisation screens of Pfs48/45 D3-B2 were set up at the Monash Macromolecular Crystallisation Platform (MMCP, Clayton, VIC, Australia). At a concentration of 6 mg/mL and at 4°C, a crystal grew in 0.2 M lithium sulphate monohydrate, 0.1 M BIS-TRIS pH 5.5, 25% (w/v) PEG 3350. Crystals were harvested with 30% glycerol in mother liquor before flash freezing in liquid nitrogen.

X-ray diffraction data was collected at the MX2 beamline of the Australian Synchrotron [67]. The XDS package [68] was used for data processing. Molecular replacement was used to solve the phase problem using structural coordinates of Pfs48/45 D3.mAgE1 (PBD ID 7UNB) and the AlphaFold2 [69] prediction of nanobody B2's structure. Iterative cycles of structure building and refinement were carried out using Coot v 0.9.8.92 [70] and Phenix v 1.21-5184 [71,72]. Interface surface area and bonds were determined by PISA v 2.1.0 [73]. Figures of the structure were prepared with PyMOL v 2.5.0 [74]. The atomic coordinates and structure factor files have been deposited in the Protein Data Bank under PDB ID 9OBN.

## *P. falciparum* maintenance and gametocyte culture

*P. falciparum* NF54 iGP2 [48] asexual parasites were cultured in human type O+ erythrocytes (Melbourne Red Cross LifeBlood) at 4% haematocrit in RPMI HEPES supplemented with 50 µg/mL hypoxanthine, 0.2% (w/v) NaHCO$_3$, and 10 mM D-Glucose) containing 5% heat-inactivated human serum and 5% Albumax (ThermoFisher Scientific). iGP2 gametocytes for transmission to mosquitoes were generated as previously described [48]. Gametocyte cultures were maintained in RPMI 1640 medium supplemented with 25.96 mM HEPES, 50 µg/mL hypoxanthine, 0.2% (w/v) NaHCO$_3$, and 10 mM D-Glucose) containing 10% heat-inactivated human serum and with daily media changes.

## Immunofluorescence assay

NF54 iGP2 stage V gametocytes were cultured as described above. Gametocytes were washed three times in warm 1X PBS. They were resuspended in warm 1X PBS and allowed to settle onto 0.1 µg/mL PHA-E (Sigma Alrich) treated coverslips and incubated in a humidified chamber at 37°C. Unbound cells were washed off using 3X 500 µL warm PBS washes. Adhered parasites were fixed by submerging in 4% paraformaldehyde in PBS (ProSciTech) for 20 min within a humidified chamber at room temperature, then washed as described above. Cells were permeabilised using 0.1% TritonX-100 (in PBS) (Thermo Scientific) within a humidified chamber at room temperature, then washed as described above. Nanobody-Fcs were diluted to 8 µg/ml in 3% BSA/PBS and incubated for 1 hr within a humidified chamber at room temperature. Coverslips were washed as described above. Goat anti-human IgG conjugated to Alexa Fluor 488 (Jackson ImmunoResearch) (diluted 1:1000 in 3% BSA/PBS) was added as the secondary antibody and incubated for 1 h within a humidified chamber. After washing, slides were incubated with anti-Pfs48/45 TB31F directly conjugated to Alexa Fluor 647 (WEHI Antibody Facility) (diluted 7.5 µg/mL in 3% BSA/PBS) for 1 h within a humidified chamber. The slides were mounted with a coverslip using VECTASHIELD PLUS antifade mounting medium with DAPI (Vector Laboratories) and sealed with nail polish. Images were acquired on a Zeiss LSM 980 using confocal mode with a 63x (1.4NA) objective lens with oil immersion using the 405, 488 and 639 nm lasers. Contrast and brightness were adjusted for visualisation using FIJI ImageJ software (v2.14.0/1.54) [75] and the Z-projection max intensity images were used for figures.

## Standard membrane feeding assays

Three- to five-day old female *Anopheles stephensi* mosquitoes were fed via a water-jacketed glass membrane feeder. Blood meals were comprised of 300 µL heat inactivated O+ human serum and 200 µL O+ erythrocytes with a stage V gametocytaemia of ~0.5%. To blood meals, 10 µL nanobody-Fcs at 5 mg/mL in PBS, 10 µL TB31F at 10 mg/mL in PBS or 10 µL PBS was added. Blood-feeding was allowed to proceed for 1–2 h, after which mosquitoes were knocked down with $CO_2$ to sort fully engorged females from unfed females or male mosquitoes. Engorged mosquitoes were transferred to tents and provided with sugar cubes and water wicks. After 7–8 days, midguts were dissected from cold-anaesthetised and ethanol sacrificed mosquitoes. Midguts were stained with 0.1% mercurochrome and oocyst numbers were counted. Data were plotted and statistical analyses were performed using GraphPad Prism 9 version 9.5.1 for Mac, GraphPad Software, Boston, Massachusetts USA, www.graphpad.com. %TRA was calculated as the percentage reduction in mean oocyst count as compared to the negative control. Statistical significance was determined using one-way ANOVA and Tukey's multiple comparison test. **** $P \leq 0.0001$. *** $P \leq 0.001$. ns $P > 0.05$.

## Supporting information

**S1 Fig. Isolation of anti-Pfs48/45 nanobodies.** (A) Purified Pfs48/45 D3 pre and post PNGase F-deglycosylation under non-reducing (NR) and reducing (R) conditions. Pre-treatment, glycosylated forms of Pfs48/45 are visible as additional bands above the expected molecular weight of 19.4 kDa. After PNGase F treatment, a single band is observed at the expected molecular weight. (B) ELISA screen of 94 phage supernatants to identify clones that are positive for Pfs48/45 D3 binding. Dashed line denotes cut-off for positive hits, defined as double the mean OD of the PBS (H1) and irrelevant nanobody negative control (H12). (C) Overview of enrichment in the 10 anti-Pfs48/45 D3 clonal groups identified by Sanger sequencing. Transmission-blocking nanobodies are highlighted in colour. The number of clones per group are indicated in brackets. (D) Line graph of the normalised counts per million (CPM) from next-generation sequencing (NGS) of the 10 anti-Pfs48/45 nanobodies across round 1 and round 2 of phage display.
(TIFF)

**S2 Fig. Anti-Pfs48/45 nanobody expression.** (A) Purified anti-Pfs48/45 nanobody-Fcs (~80 kDa) under non-reducing and reducing conditions. (B) Purified monomeric anti-Pfs48/45 nanobodies (~15 kDa) under non-reducing and reducing conditions.
(TIFF)

**S3 Fig. Affinity curves for nanobody-Fc binding to Pfs48/45 D3 and interactions between Pfs48/45 and nanobody B2.** (A) Representative binding curves of different concentrations of Pfs48/45 D3 to immobilised nanobody-Fcs and TB31F. Binding curves were generated by bio-layer interferometry and curves were fitted using a 1:1 Langmuir binding model. Binding affinities ($K_D$) are indicated above binding curves. (B) Close-up view of the crystal structure of the Pfs48/45- nanobody B2 complex showing hydrogen bonds and ionic interactions between them. Ribbon representation of Pfs48/45 D3 in yellow and nanobody B2 in blue.
(TIFF)

**S1 Table. Anti-Pfs48/45 nanobody (Nb) amino acid sequences.**
(DOCX)

**S2 Table. Summary of interactions between Pfs48/45 D3 and nanobody B2.**
(DOCX)

## Acknowledgments

We thank Cindy Luo from the WEHI and Geoffrey Kong from the Monash Macromolecular Crystallisation Platform (MMCP, Clayton, VIC, Australia) for assistance with setting up the crystallisation screens. We thank Matthew T. O'Neill, Stephanie Trickey and Sravya Keremane for assistance with initial cloning and recombinant protein expression. This research was undertaken using the MX2 beamline at the Australian Synchrotron, part of ANSTO, and made use of the Australian Cancer Research Foundation (ACRF) detector. We thank the MX2 beamline staff at the Australian Synchrotron for their assistance during data collection.

## Author contributions

**Conceptualization:** Melanie H. Dietrich, Wai-Hong Tham.

**Formal analysis:** Frankie M. T. Lyons, Jill Chmielewski, Melanie H. Dietrich, Wai-Hong Tham.

**Funding acquisition:** Wai-Hong Tham.

**Investigation:** Frankie M. T. Lyons, Jill Chmielewski, Mikha Gabriela, Li-Jin Chan, Joshua Tong, Amy Adair, Kathleen Zeglinski, Quentin Gouil, Melanie H. Dietrich.

**Methodology:** Frankie M. T. Lyons, Jill Chmielewski.

**Supervision:** Melanie H. Dietrich, Wai-Hong Tham.

**Visualization:** Frankie M. T. Lyons, Kathleen Zeglinski, Wai-Hong Tham.

**Writing – original draft:** Frankie M. T. Lyons, Melanie H. Dietrich, Wai-Hong Tham.

**Writing – review & editing:** Frankie M. T. Lyons, Jill Chmielewski, Mikha Gabriela, Li-Jin Chan, Amy Adair, Kathleen Zeglinski, Quentin Gouil, Melanie H. Dietrich, Wai-Hong Tham.

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
