## [Decision Letter · Decision Letter 0]

28 Aug 2025

Pfs48/45 nanobodies block Plasmodium falciparum transmission

PLOS Pathogens

Dear Dr. *Wai-Hong Tham* ,

Please submit your revised manuscript within 60 days . If you will need more time than this to complete your revisions, please reply to this message or contact the journal office at plospathogens@plos.org. Please include the following items when submitting your revised manuscript:

We look forward to receiving your revised manuscript.

Kind regards,

Zbynek Bozdech

Guest Editor

PLOS Pathogens

Tracey Lamb

Section Editor

Editor-in-Chief

PLOS Pathogens

Michael Malim

Editor-in-Chief

PLOS Pathogens

orcid.org/0000-0002-7699-2064

**Journal Requirements:**

- ® on page: 24 and 41.

4) Tables should not be uploaded as individual files. Please remove these files and include the Tables in your manuscript file as editable, cell-based objects. For more information about how to format tables, see our guidelines:

https://journals.plos.org/plospathogens/s/tables

5) Please ensure that the funders and grant numbers match between the Financial Disclosure field and the Funding Information tab in your submission form. Note that the funders must be provided in the same order in both places as well.

**Reviewer #1**

**Reviewers' Comments:**

Reviewer's Responses to Questions

**Part I - Summary**

Reviewer #1: This is a well-conducted study in which the authors develop a panel of nanobodies targeting the transmission-blocking malaria vaccine candidate Pfs48/45. They find that some of these nanobodies can block the transmission of parasites to mosquitos. They determine the crystal structure of the best of these nanobodies bound to the D3 domain of Pfs48/45 and show that overlaps with the epitope of some, but not all transmission-blocking antibodies against domain D3. This study is of interest and, as explained in the discussion, the nanobodies might have some applications in addition to those found for the antibodies. It is therefore a worth-while study.

The second part of the study combines the best nanobody from this study with a nanobody targeting a second transmission blocking vaccine immunogen Pfs230 as a bispecific. This is tested at single concentration and is as effective at blocking transmission as the two parental nanobodies. I found this part of the study quite hard to interpret as the concentration dependence of the bispecific was not compared with that of the monospecifics with TRAs just done at one concentration at which there was very high TRA. It is therefore hard to conclude much from this part of the study. Line 288 stated that “bispecific nanobodies do not appear to have significantly higher transmission-blocking activity that B2-Fc” and I did not agree that the authors have the evidence for this statement either way. It is also the case that these were not compared with a monovalent nanobody and so it also can’t be concluded that the two arms were both functional. More data is required here to draw clear conclusions from this finding.

In general therefore, the first part of Pfs48/45 nAbs is clear and well done, but the section on the bispecific is hard to interpret and is less well controlled. The paper might be better without the bispecific section, allowing the authors to conduct a fully controlled study on these bispecifics.

**Part II – Major Issues: Key Experiments Required for Acceptance**

Reviewer #1: No experiments for the Pfs48/45 nanobody part.

For the bispecific, if this remains in the paper, it would be best to conduct the TRAs at different dilutions to know how it performs relative to monospecific and Fabs.

**Part III – Minor Issues: Editorial and Data Presentation Modifications**

Reviewer #1: (No Response)

PLOS authors have the option to publish the peer review history of their article (what does this mean? ). If published, this will include your full peer review and any attached files.

**Do you want your identity to be public for this peer review?** For information about this choice, including consent withdrawal, please see our Privacy Policy .

Reviewer #1: No

**Reviewer #2**

In this manuscript the authors have explored the feasibility of developing nanobodies

against the sexual stage antigens, Pfs48/45 and Pfs230, to inhibit transmission of P.

falciparum. Following immunization of alpacas with deglycosylated Pfs48/45 D3 a

nanobody phage library was generated and screened for binding to Pfs48/45 by ELISA. Anti-

Pfs48/45 nanobodies were produced as recombinant proteins fused to Fc and tested for

recognition of Pfs48/45 by ELISA and immunofluorescence assay (IFA). The interaction of

three nano-Fcs, A4-Fc, B2-Fc and C3-Fc, that bind Pfs48/45 D3 domain was analyzed. The

three nanobodies react to overlapping epitopes on Pfs48/45 D3 but don’t inhibit binding of

a previously identified mAb referred to as TB31F. The three nanobodies, A4-Fc, B2-Fc and

C3-Fc, like TB31F, block parasite transmission with high eYiciency. The binding site of one

of the nanobodies, B2-Fc, was determined by solving the structure of the nanobody in a

complex with Pfs48/45 D3. By linking previously identified nanobodies, F5 and F10, against

Pfs230, with B2 the authors produced chimeric bispecific nanobodies, that react with both

Pfs48/45 and Pfs230. Importantly, the chimeras retain the ability to target the mosquito

stages and block transmission eYiciently. The authors have shown elegantly the feasibility

of using nanobodies to inhibit P. falciparum transmission. They should address the

following main comment:

The characterization of the nanobodies in terms of binding to the antigens and mapping of

the binding epitopes is clearly described. However, the function studies testing the ability

of the nanobodies to block parasite transmission need additional work. The nanobodies

were tested for inhibition of transmission at a single dilution at which close to 100%

inhibition is achieved. It would have been informative to test the nano-Fcs at diYerent

concentrations to identify the most potent nano-bodies for transmission blocking activity.

The experiment with bispecific nano-Fcs was also performed at a single concentration. At

the concentration used even mono-specific nanobodies yield close to 100% inhibition. It is

therefore diYicult to determine if there is any added value to using bi-specific nanobodies.

Instead of testing the nano-Fcs at a concentration where all of them yield close to 100%

inhibition, it may be useful to test the nano-Fcs at a range of concentrations to allow

determination of an IC50. This may allow comparison of the potency of diYerent nano-Fcs

including the chimeric nanobodies and determine if the bi-specific nanobodies are more

potent that mono-specific nanobodies.

**Figure resubmission:**
---

## [Editor Report · Decision Letter 1]

16 Sep 2025

Pfs48/45 nanobodies block Plasmodium falciparum transmission

PLOS Pathogens

Dear Dr. Tham,

Thank you for submitting your revised manuscript to PLOS Pathogens. It has come to our attention that not all the reviewers were included in the previous decision letter due to a technical error. Please accept our sincere apologies for any inconvenience. In light of this, we are resending the decision with all reviews included, and invite you to submit a revised version of the manuscript that addresses the points raised during the review process.

Please submit your revised manuscript within 60 days Nov 15 2025 11:59PM. If you will need more time than this to complete your revisions, please reply to this message or contact the journal office at plospathogens@plos.org. Please include the following items when submitting your revised manuscript:

We look forward to receiving your revised manuscript.

Kind regards,

Zbynek Bozdech

Guest Editor

PLOS Pathogens

Tracey Lamb

Section Editor

PLOS Pathogens

Editor-in-Chief

PLOS Pathogens

orcid.org/0000-0003-2946-9497

Editor-in-Chief

PLOS Pathogens

orcid.org/0000-0002-7699-2064

**Reviewers' Comments:**

Reviewer's Responses to Questions

**Part I - Summary**

**Reviewer #1:**  This is a well-conducted study in which the authors develop a panel of nanobodies targeting the transmission-blocking malaria vaccine candidate Pfs48/45. They find that some of these nanobodies can block the transmission of parasites to mosquitos. They determine the crystal structure of the best of these nanobodies bound to the D3 domain of Pfs48/45 and show that overlaps with the epitope of some, but not all transmission-blocking antibodies against domain D3. This study is of interest and, as explained in the discussion, the nanobodies might have some applications in addition to those found for the antibodies. It is therefore a worth-while study.

The second part of the study combines the best nanobody from this study with a nanobody targeting a second transmission blocking vaccine immunogen Pfs230 as a bispecific. This is tested at single concentration and is as effective at blocking transmission as the two parental nanobodies. I found this part of the study quite hard to interpret as the concentration dependence of the bispecific was not compared with that of the monospecifics with TRAs just done at one concentration at which there was very high TRA. It is therefore hard to conclude much from this part of the study. Line 288 stated that “bispecific nanobodies do not appear to have significantly higher transmission-blocking activity that B2-Fc” and I did not agree that the authors have the evidence for this statement either way. It is also the case that these were not compared with a monovalent nanobody and so it also can’t be concluded that the two arms were both functional. More data is required here to draw clear conclusions from this finding.

In general therefore, the first part of Pfs48/45 nAbs is clear and well done, but the section on the bispecific is hard to interpret and is less well controlled. The paper might be better without the bispecific section, allowing the authors to conduct a fully controlled study on these bispecifics.

**Reviewer #2:**

Lyons et al. develop the first set of nanobodies against the Plasmodium falciparum surface protein Pfs48/45, one of the leading targets for transmission-blocking vaccine development. They identify a set of three nanobodies that bind to Pfs48/45 domain 3 with nanomolar affinities and all bind the same epitope. They assess their transmission-blocking potential in standard membrane feedings assays (SMFAs), showing that these nanobodies exhibit high TRA at 100ug/mL. However, since the nanobodies were only tested at a single high dose, it remains unclear how potent these antibodies really are and thus compare to previously reported monoclonal antibodies. Furthermore, they solved the structure of the most potent nanobody B2 in complex with Pfs48/45 domain 3 allowing them to identify the B2 epitope. This B2 epitope largely overlaps with the previously reported epitope of potent human antibodies RUPA-44 and RUPA-117. The authors continue with the most promising nanobody to develop two bispecific nanobodies, that combine nanobody B2 with variable domains of nanobodies targeting TBV candidate Pfs230D1. They show that these bispecific nanobodies can still bind both targets and block transmission at 100 ug/mL. However, the design of their SMFA experiment could not show how the potency of the bi-specific nanobodies compare to the parental nanobodies and thus whether there is synergistic/additive/no/negative effect when combining two specificities in one molecule.

The novelty of this study is that it describes the first nanobodies against Pfs48/45. These may indeed be useful tools for studying biology and generating parasite-resistant mosquitoes, though it is more difficult to see how these (non-human) nanobodies can become attractive for prophylactic use in humans in the near future. Another novelty described in this study is the generation and testing of bi-specific antibodies against TBV candidates, but more functional assays are required to assess their value.

The isolation, binding assessment and structural analyses of the nanobodies are well executed and described. The major weakness of this study is that functional characterisation is very limited and provides little insight in the actual potency of (bi-specific) nanobodies. More in-depth functional analysis is required to understand the potential value of these nanobodies, see below.

**Reviewer #3:**  Malaria transmission-blocking vaccines (TBVs) are crucial strategies for eliminating malaria, with Pfs48/45 and Pfs230 being the primary vaccine targets. The authors developed nanobodies against Pfs48/45 that recognize gametocytes and exhibit strong transmission-reducing activity. The crystal structure of the most potent nanobody bound to the recombinant Pfs48/45 D3 revealed that it attaches to a different epitope than the known human mAb, TB31F. Importantly, they also created bispecific nanobodies capable of targeting both Pfs48/45 and Pfs230, fused to a human Fc domain. These bispecific nanobodies recognize both Pfs48/45 and Pfs230, thereby reducing malaria transmission.

Very interesting and important work. Establishing bispecific nanobodies targeting both Pfs48/45 and Pfs230 could serve as an alternative approach to current TBV development efforts, supporting malaria elimination. All experiments are carefully designed and carried out to a high standard by leading experts in this field of research. I have some comments to further improve this manuscript.

**Part II – Major Issues: Key Experiments Required for Acceptance**

**Reviewer #1:**  No experiments for the Pfs48/45 nanobody part.

For the bispecific, if this remains in the paper, it would be best to conduct the TRAs at different dilutions to know how it performs relative to monospecific and Fabs.

**Reviewer #2** : • Figure 2: The functional assessment of the nanobodies should be expanded with titrations of the nanobodies. The nanobodies should be diluted till TRA is lost or drops below 50-80% to allow estimation of their IC80 value and comparison with other antibodies described in literature.

• Figure 4: The bispecific nanobodies show very high TRA, but so does the parental nanobody B2-Fc. The TRA is almost saturated making it impossible to compare the potency of the (bi-specific) nanobodies. Side-by-side titrations of the (parental) nanobodies and bispecific nanobodies should be conducted to be able to assess whether the bispecific nanobodies are more potent, similarly potent, or less potent than their parental counterparts. Note that the parental nanobodies F5 and F10 should also be included in these experiments. If bispecific nanobodies turn out to be more potent than their parental counterparts, the authors could test whether this can be explained by assessing whether bispecifics have a higher apparent affinity to gametes.

**Reviewer #3:** 1) Line 189, Figure 2B

The co-localization of TB31F and three nanobodies’ staining is clear. However, the lattice-like staining pattern was unusual because Pfs48/45 is located on the surface of gametocytes or gametes. I believe that the membrane structure of the gametocyte antigen was damaged during sample preparation, including fixation. Therefore, I strongly recommend the authors repeat the IFA experiment using another fixation method, such as PFA fixation, which is commonly used in other studies.

2) Lines 255-264, SMFA of bispecific nanobody-Fcs, Figure 4C

It is crucial to compare the potency of mono- and bispecific nanobodies. Therefore, I strongly recommend that the authors add dose-response SMFA to calculate the IC50. Then they can compare the potencies.

3) Lines 1000-1001, S1 Fig A

Some additional bands are still visible in the post-treatment lanes. Please estimate the purity by densitometry.

**Part III – Minor Issues: Editorial and Data Presentation Modifications**

**Reviewer #1:**  (No Response)

**Reviewer #2:**  • Line 27: “…reveals it binds a distinct epitope to TB31F, a leading transmission-blocking monoclonal antibody.” This phrasing suggests that the nanobody B2 binds a novel epitope on the Pfs48/45 surface, while it largely overlaps with the previously characterized RUPA-44 and RUPA-117 epitope. Please refer to the RUPA-44/RUPA-117 epitope in the abstract.

• Line 73: Please mention the extended conformation observed by cryo-EM as observed by Kucharska et al. (2025).

• Line 86: In the same paper, by Kucharska et al., the authors have characterized the epitopes of mAbs RUPA-58 and RUPA-154, which bind to D1 and D2, respectively. Include these in the description here.

• Lines 156-160: Can the authors speculate why some nanobodies show binding by Western Blot and ELISA, but not by BLI?

• Line 160: Figure S3 mentions Kd = 3.3 nM, not 3.8 nM.

• Line 212-215: Please include a supplementary figure showing the hydrogen/ionic interactions between Pfs48/45 D3 and B2

• Figure 3C: it would be more meaningful to replace the figure with the non-overlapping RUPA-29 epitope with a figure showing the RUPA-117 epitope, which largely overlaps with B2 and RUPA-44.

• Figure 4B: the mentioned affinities do not correspond to the ones found in the text or Figure S4C.

• Line 253-254: Please change Pfs230 to Pfs230 D1 and Pfs48/45 to Pfs48/45 D3.

• Line 286: We agree that based on the reported structures, co-binding on the same complex is unlikely. However, since Pfs48/45 and Pfs230 are highly expressed, it does not seem impossible for the same nanobody to bind two different complexes at the same time.

• Line 288: “… which may explain why the bispecific nanobodies do not appear to have a signficantly higher transmission-blocking activity.” Please update this statement following the outcome of the requested SMFA titration experiments.

• Line 305 – 318: the supposed advantages of using a nanobody compared to the already humanized TB31F or one of the previously characterized human-derived monoclonal antibodies, which would not suffer from the mentioned setbacks, is not made clear in the context of clinical use.

• Line 328: There are more examples of expressing single chain monoclonal antibodies against mosquito-stages, including Green et al. (2023) which use anti-CSP antibody sc2A10 to block transmission by targeting hemolymph sporozoites.

• Line 583: it would be helpful to explicitely mention that human serum contained active human complement (assuming it did).

• Figure legends 2d and 4c: please include antibody/nanobody concentrations in the figure legends and specify whether complement was added.

**Reviewer #3** : 1) Line 318 “humanisation of the nanobody backbone [55] would need to be considered.”

I agree with this statement because this step is crucial for developing this bispecific new intervention to move toward clinical development. When I read reference #55, humanization of camelid Nanobodies is possible but not straightforward. Please cite a reference that details the clinical development of nanobody-based human antibodies (as an example) if available.

2) Recombinant Pfs48/45 D3 expression

Why did the Authors use two different methodologies to express “Pfs48/45 D3 (Sf21)” and “stabilized Pfs48/45 D3 (Expi293)”? Please explain.

3) Others:

Line 283: “in two preprints” -> The Author’s group paper has been published in Science recently.

Line 393: “GERBU FAMA” -> Please add company information.

Line 537: “molar ratio of 1:1.5” -> Why not 1:1? Please explain.

Line 586: “Blood-feeding….for 1-2 h” -> Why so long?

Usually, the mosquitoes become full-stomach in 30 minutes.

Line 948: Figure 1G

Why are non-competitive wells higher than 100%? Please explain.

PLOS authors have the option to publish the peer review history of their article (what does this mean? ). If published, this will include your full peer review and any attached files.

**Do you want your identity to be public for this peer review?** For information about this choice, including consent withdrawal, please see our Privacy Policy .

Reviewer #1: No

Reviewer #2: No

Reviewer #3: No

**Figure resubmission:**

**Reproducibility:**



---

## [Decision Letter · Decision Letter 2]

24 Dec 2025

Pfs48/45 nanobodies block Plasmodium falciparum transmission

PLOS Pathogens

Dear Dr. Tham,

Thank you for submitting your manuscript to PLOS Pathogens. After careful consideration, we feel that it has merit but does not fully meet PLOS Pathogens's publication criteria as it currently stands. Therefore, we invite you to submit a revised version of the manuscript that addresses the points raised during the review process.

We look forward to receiving your revised manuscript.

Kind regards,

Tracey J. Lamb

Section Editor

PLOS Pathogens

Tracey Lamb

Section Editor

PLOS Pathogens

Editor-in-Chief

PLOS Pathogens

orcid.org/0000-0003-2946-9497

Michael Malim

Editor-in-Chief

PLOS Pathogens

orcid.org/0000-0002-7699-2064

**Additional Editor Comments:**

Please address the comments of reviewer 1- summary section.

**Journal Requirements:**

**Reviewers' Comments:**

Reviewer's Responses to Questions

**Part I - Summary**

Reviewer #1: This is a tricky one. All three reviewers share the view that the authors should do a dose response titration for their bispecific antibody to show if they are or are not an improvement on the monospecific antibodies.

Reviewer 1: "it would be best to conduct the TRAs at different dilation"

Reviewer 2: "more in depth functional analysis is required to understand the potential value of these (bispecific) nanobodies"

Reviewer 3 " it is crucial to compare the potency....I recommend that the authors add dose-response SMFA data'

But the authors assert that this is outside the scope of the study as there was never an intention that the bispecific would be better.

Personally I don't understand the point of making a bispecific, which is going to bind to both antigens if the technical part is done correctly, without testing if it is better than monospecific. They have made the reagent, but not determined whether it is useful. I don't understand how they can assert that testing if it is better or not is outside the scope of the study. Surely this is the entire point of the study??

Personally I would support removing the bispecific and publishing the nanobody study, which is well done and suitable for publication. I would be recommending "accept" for that. They could then do the dose titration for the bispecific and publish a second paper on it.

I suppose, and I do generally support the right of authors to publish what they want if correctly done, that if they are determined, they could include the bispecific production. But then they should be very explicit in the paper that this has not been tested to the degree that they know if it is better than the monospecific and therefore can't make any comment on whether it is an improvement. Although from my perspective, this would seem unusual.

Reviewer #2: I would like to thank the authors for addressing some of my comments, though the major ones still need to be addressed, see below.

Reviewer #3: Although all the reviewers commented on the dose response SMFA, the Authors' rebuttal is reasonable.

I think the other comments are appropriately addressed.

**Part II – Major Issues: Key Experiments Required for Acceptance**

Reviewer #1: (No Response)

Reviewer #2: I strongly disagree with the authors on their suggestion that titrations are not needed. They present their newly discovered and developed nanobodies as versatile antibodies that can reduce malaria transmission (also as concluding sentence of their abstract). It will therefore be important to titrate the nanobodies to understand (i) how potent these new nanobodies actually are and how that roughly compares to the many (human) monoclonal antibodies that have been described for Pfs48/45 so far and (ii) whether their bispecific nanobodies are less, equally or even more potent than the corresponding monospecific nanobodies.

It will be frustrating for readers to not have this extra information that can readily be generated. I understand that SMFAs are labor intensive, but the number of experiments that would be needed to address the points above does not need to be extensive. Note that rough estimates of potencies will be acceptable, e.g. understanding whether activity drops below 80% somewhere between xx and yy ug/mL.

The argument that many other papers did not include titrations does in my opinion not hold since Tang et al. included a titration of their most potent mAbs in the supplementary data S7, Miura et al. tested mAbs with no activity (so why titrate?) and the other papers used the mAbs to identify new potent epitopes than presenting the mAbs as transmission blocking tools.

Altogether, I strongly believe that additional titrations are needed and am supported in this by the other reviewers. These extra data, irrespective of the outcome, will certainly strengthen the paper.

Reviewer #3: Not required.

**Part III – Minor Issues: Editorial and Data Presentation Modifications**

Reviewer #1: (No Response)

Reviewer #2: Most of my minor comments have been addressed, thank you. I would just like to suggest to present the nanobody concentrations in the SMFA figure legends, instead of the absolute amounts. This will help the readers to interpret the data more quickly, rather than having to do the calculations themselves.

Reviewer #3: No

PLOS authors have the option to publish the peer review history of their article (what does this mean? ). If published, this will include your full peer review and any attached files.

**Do you want your identity to be public for this peer review?** For information about this choice, including consent withdrawal, please see our Privacy Policy .

Reviewer #1: No

Reviewer #2: No

Reviewer #3: No

**Figure resubmission:**

**Reproducibility:**



---

## [Editor Report · Decision Letter 3]

9 Jan 2026

Dear Prof Tham,

We are pleased to inform you that your manuscript 'Pfs48/45 nanobodies block Plasmodium falciparum transmission' has been provisionally accepted for publication in PLOS Pathogens.

Best regards,

Tracey J. Lamb

Section Editor

PLOS Pathogens

Tracey Lamb

Section Editor

PLOS Pathogens

Sumita Bhaduri-McIntosh

Editor-in-Chief

PLOS Pathogens

orcid.org/0000-0003-2946-9497

Michael Malim

Editor-in-Chief

PLOS Pathogens

orcid.org/0000-0002-7699-2064
---

## [Editor Report · Acceptance letter]

Dear Prof Tham,

We are delighted to inform you that your manuscript, "Pfs48/45 nanobodies block Plasmodium falciparum transmission," has been formally accepted for publication in PLOS Pathogens.

Best regards,

Sumita Bhaduri-McIntosh

Editor-in-Chief

PLOS Pathogens

orcid.org/0000-0003-2946-9497

Michael Malim

Editor-in-Chief

PLOS Pathogens

orcid.org/0000-0002-7699-2064